# Electrophysiological Responses of *Bactrocera kraussi* (Hardy) (Tephritidae) to Rectal Gland Secretions and Headspace Volatiles Emitted by Conspecific Males and Females

**DOI:** 10.3390/molecules26165024

**Published:** 2021-08-19

**Authors:** Sally Noushini, Soo Jean Park, Jeanneth Perez, Danielle Holgate, Vivian Mendez, Ian M. Jamie, Joanne F. Jamie, Phillip W. Taylor

**Affiliations:** 1Department of Molecular Sciences, Macquarie University, Sydney, NSW 2109, Australia; danielle.holgate@hdr.mq.edu.au (D.H.); ian.jamie@mq.edu.au (I.M.J.); joanne.jamie@mq.edu.au (J.F.J.); 2Australian Research Council Industrial Transformation Training Centre for Fruit Fly Biosecurity Innovation, Macquarie University, Sydney, NSW 2109, Australia; soojean.park@mq.edu.au (S.J.P.); jeanneth.perez@gmail.com (J.P.); vivian.mendez@mq.edu.au (V.M.); phil.taylor@mq.edu.au (P.W.T.); 3Applied BioSciences, Macquarie University, Sydney, NSW 2109, Australia

**Keywords:** Tephritidae, pheromone, olfaction, electrophysiology, GC–MS, EAD, EPD

## Abstract

Pheromones are biologically important in fruit fly mating systems, and also have potential applications as attractants or mating disrupters for pest management. *Bactrocera kraussi* (Hardy) (Diptera: Tephritidae) is a polyphagous pest fruit fly for which the chemical profile of rectal glands is available for males but not for females. There have been no studies of the volatile emissions of either sex or of electrophysiological responses to these compounds. The present study (i) establishes the chemical profiles of rectal gland contents and volatiles emitted by both sexes of *B. kraussi* by gas chromatography–mass spectrometry (GC–MS) and (ii) evaluates the detection of the identified compounds by gas chromatography–electroantennogram detection (GC–EAD) and –electropalpogram detection (GC–EPD). Sixteen compounds are identified in the rectal glands of male *B. kraussi* and 29 compounds are identified in the rectal glands of females. Of these compounds, 5 were detected in the headspace of males and 13 were detected in the headspace of females. GC–EPD assays recorded strong signals in both sexes against (*E*,*E*)-2,8-dimethyl-1,7-dioxaspiro[5.5]undecane, 2-ethyl-7-mehtyl-1,6-dioxaspiro[4.5]decane isomer 2, (*E*,*Z*)/(*Z*,*E*)-2,8-dimethyl-1,7-dioxaspiro[5.5]undecane, and (*Z*,*Z*)-2,8-dimethyl-1,7-dioxaspiro[5.5]undecane. Male antennae responded to (*E*,*E*)-2,8-dimethyl-1,7-dioxaspiro[5.5]undecane, 2-methyl-6-pentyl-3,4-dihydro-2*H*-pyran, 6-hexyl-2-methyl-3,4-dihydro-2*H*-pyran, 6-oxononan-1-ol, ethyl dodecanoate, ethyl tetradecanoate and ethyl (*Z*)-hexadec-9-enoate, whereas female antennae responded to (*E*,*E*)-2,8-dimethyl-1,7-dioxaspiro[5.5]undecane and 2-methyl-6-pentyl-3,4-dihydro-2*H*-pyran only. These compounds are candidates as pheromones mediating sexual interactions in *B. kraussi*.

## 1. Introduction

Chemical communication plays an important role in the mating behaviour of tephritid fruit flies [1,2]. Volatile compounds, typically stored in the rectal glands and emitted into the air during calling and courtship, commonly attract members of the opposite sex [3,4,5], as well as members of the same sex to mating aggregations [6,7]. Studies of the volatile compounds emitted by dacine fruit flies have focused on the chemical profiles of males because they have been classically considered as the major sex pheromone producers [8]. However, in some species, females have been found to produce sex pheromones. For example, in *Bactrocera oleae*, sex pheromones are produced by female flies [9,10], while males produce a compound that only acts as a close-range attractant for females [11,12]. Females of *Rhagoletis coversa* and *R. brncici* deposit fruit-marking pheromones that also attract males for mating [13]. It is likely that the historical focus on male-produced compounds has led to an underestimation of the importance of female-produced compounds in fruit fly chemical ecology; both sexes need to be studied for a comprehensive understanding of chemical communication.

*Bactrocera* and *Zeugodacus* fruit flies are known to produce and emit diverse compounds, including aliphatic amides, spiroacetals, alcohols, pyrazines, dihydropyrans and esters of medium chain fatty acids [8,14,15,16,17,18,19,20,21,22,23]. Although less common, aromatic compounds are also found in *Zeugodacus cucurbitae* [15]. The most commonly reported compound in *Bactrocera* and *Zeugodacus* rectal gland secretions is (*E*,*E*)-2,8-dimethyl-1,7-dioxaspiro[5.5]undecane [8,15,16,19,20,22,23,24,25,26]. This compound is also known to occur in other insects, such as *Polistes* wasps and *Ontholestes* beetles [27,28]. *N-*(3-Methylbutyl)acetamide is a major component in *Bactrocera tryoni* male rectal glands [14,23] and is also widely reported in other *Bactrocera* and *Zeugodacus* species [15,16,22,23,25,26]. Female *Bactrocera* and *Zeugodacus* have been found to produce the esters of medium chain fatty acids [8,15,22,23,26].

*Bactrocera kraussi* (Hardy) is endemic to the Torres Strait Islands and Northeast Queensland, as far south as Townsville, Australia [29]. *Bactrocera kraussi* is a polyphagous and economically important pest species, with a wide range of both wild and commercial hosts, including mango (*Mangifera indica*), banana (*Musa* spp.), guava (*Psidium guajava*), feijoa (*Acca sellowiana*), peach (*Prunus persica*), diverse citrus species, and tamarind (*Tamarindus indica*) [29,30]. The compounds stored in the rectal glands of *B. kraussi* males have been partially described by Fletcher et al. (1992), and included (*E*,*E*)-2,8-dimethyl-1,7-dioxaspiro[5.5]undecane as a major compound, its (*Z*,*Z*)-isomer, diastereomer(s) of 2-ethyl-7-methyl-1,6-dioxaspiro[4.5]decane, (*E*,*E*)-2-ethyl-8-methyl-1,7-dioxaspiro-[5.5]undecane, 2,8-dimethyl-1,7-dioxaspiro[5.5]undecan-3-ol, *N-*(2-methylbutyl)acetamide, *N-*(3-methylbutyl)acetamide, diethyl butanedioate, 2-methyl-6-pentyl-3,4-dihydro-2*H*-pyran, 3-methylbutan-1-ol, 6-oxoundecan-2-ol and 6-oxononan-1-ol [19]. The compounds stored in the rectal glands of *B. kraussi* females are unknown. Further, there have been no studies of the headspace emissions of male or female *B. kraussi*, or of their potential biological significance. The present study (i) describes the chemical profiles of rectal glands and headspace in males and females of *B. kraussi*, including re-description of male rectal gland contents, by using gas chromatography–mass spectrometry (GC–MS) and (ii) evaluates the antennal and palpal responses to each compound in natural blends by using gas chromatography–electroantennogram detection (GC–EAD) and –electropalpogram detection (GC–EPD).

## 2. Results

### 2.1. Analysis of Rectal Gland Extracts and Headspace Collections

The compositions of *B. kraussi* rectal gland extracts and volatile emissions from sexually mature males and females are summarized in Table 1. Chromatograms of both the rectal gland and headspace of both sexes are shown in Figure 1. Spiroacetals were the dominant class of compounds in male rectal glands, whereas esters of saturated/unsaturated fatty acids were the main class of compounds in female rectal glands. Eleven of the 13 previously identified compounds have been confirmed. Additionally, 2,7-dimethyl-1,6-dioxaspiro[4.5]decane, (*E*,*Z*)/(*Z*,*E*)-2,8-dimethyl-1,7-dioxaspiro[5.5]undecane, another isomer of 2-ethyl-7-methyl-1,6-dioxaspiro[4.5]decane, 2-methyl-8-propyl-1,7-dioxaspiro[5.5]undecane, 2-ethyl-1-hexanol and 6-hexyl-2-methyl-3,4-dihydro-2*H-*pyran have been identified. Only five of the rectal gland compounds were detected in male headspace samples. The most abundant compound in both male rectal glands and headspace samples was (*E*,*E*)-2,8-dimethyl-1,7-dioxaspiro[5.5]undecane (**5**), representing about 85.6% and 45.1% of the blends, respectively. *N*-(3-Methylbutyl)acetamide (**4**) was found as a minor compound (2.2%) in male rectal glands but appeared as a major compound (44.3%) in the headspace samples. A total of 29 compounds were identified in *B. kraussi* female rectal glands, including all the spiroacetals found in males, except 2-ethyl-7-methyl-1,6-dioxaspiro[4.5]decane isomer 2, *N-*(3-methylbutyl)acetamide, and the 20 esters of saturated/unsaturated fatty acids (Table 1). The spiroacetals (*E*,*E*)-2,8-dimethyl-1,7-dioxaspiro[5.5]undecane (**5**), 2-ethyl-7-methyl-1,6-dioxaspiro[4.5]decane isomer 1 (**6**) and (*E*,*E*)-2-ethyl-8-methyl-1,7-dioxaspiro[5.5]undecane (**10**) are in noticeably smaller amounts than in males. Of these, 13 compounds were detected in headspace samples. The predominant compound present in female gland extracts and headspace samples was ethyl dodecanoate (**19**), although it was found in higher proportions in the headspace samples (70.2% vs. 39.1%). The second major compound, ethyl tetradecanoate (**25**), had similar relative abundance in rectal gland extracts and headspace samples (25.6% and 23.7%, respectively).

### 2.2. Electrophysiology

Electroantennographic (EAD) and electropalpographic detection (EPD) of male and female *B. kraussi* to the rectal gland extracts of conspecific males and females are presented in Figure 2. Overall, EPD detected stronger responses to the endogenous compounds than EAD did under the experimental conditions. Maxillary palps of both sexes produced strong signals corresponding to (*E*,*E*)-2,8-dimethyl-1,7-dioxaspiro[5.5]undecane (**5**), its two diastereomers (**9**, **13**) and 2-ethyl-7-methyl-1,6-dioxaspiro[4.5]decane isomer 2 (**11**) (Figure 2). Male antennae produced signals corresponding to (*E*,*E*)-2,8-dimethyl-1,7-dioxaspiro[5.5]undecane (**5**), the two dihydropyrans (**8**, **13**), 6-oxononan-1-ol (**16**), ethyl dodecanoate (**19**), ethyl tetradecanoate (**25**) and ethyl (*Z*)-hexadec-9-enoate (**31**), while female antennae produced signals corresponding to only (*E*,*E*)-2,8-dimethyl-1,7-dioxaspiro[5.5]undecane (**5**) and 2-methyl-6-pentyl-3,4-dihydro-2H-pyran (**8**). (*E*,*E*)-2,8-Dimethyl-1,7-dioxaspiro[5.5]undecane (**2**) elicited responses from both antennae and palps of both sexes.

## 3. Discussion

This paper is the first to describe the chemical profiles of female *B. kraussi* rectal glands and headspace, and to describe the chemical profiles of male *B. kraussi* headspace. It is also the first report of electrophysiological detection to rectal gland volatiles in male and female *B. kraussi*. The combined analysis of rectal gland extracts and headspace (volatile) collections with GC–EAD/EPD provides a robust starting point for understanding the chemical communication of *B. kraussi*.

The present study confirms the presence and identity of compounds that have been previously reported in rectal glands of *B. kraussi* males, except 3-methylbutan-1-ol and 2-hydroxyundecan-6-one [19] that were not detected in the present study, and identifies additional six compounds. The previous study synthesized all the diastereomers of 2,8-dimethyl-1,7-dioxaspiro[5.5]undecane to confirm which stereoisomers were present but reported that only (*E*,*E*)- (**5**) and (*Z*,*Z*)- (**12**) isomers co-occur in *B. kraussi* males [19]. In the present study, another peak was observed at an earlier retention time than (*Z*,*Z*)- isomer (**12**), which exhibited very similar mass fragmentation patterns to another isomer (**9**) (Table 1). The diastereomer (**9**) is presumably the (*E*,*Z*)- or (*Z*,*E*)-isomer of 2,8-dimethyl-1,7-dioxaspiro[5.5]undecane. The other new spiroacetals include 2,7-dimethyl-1,6-dioxaspiro[4.5]decane (**2**), diastereomer(s) (**11**) of 2-ethyl-7-methyl-1,6-dioxaspiro[4.5]decane, and 2-methyl-8-propyl-1,7-dioxaspiro[5.5]undecane (**14**). The spiroacetal 2,7-dimethyl-1,6-dioxaspiro[4.5]decane (**2**) was assigned by examining an apparent molecular ion of *m*/*z* 170 and the fragment ions at *m*/*z* 101 and 98, indicative of a methyl substituted five-membered ring. The overall fragmentation patterns of 2,7-dimethyl-1,6-dioxaspiro[4.5]decane (**2**) are consistent with previous findings [20]. Fletcher et al. (1992) reported a peak corresponding to the (*E*,*E*)- or (*E*,*Z*)-2-ethyl-7-methyl-1,6-dioxaspiro[4.5]decane (**11**) [19], but the present study observed another peak with very similar mass fragmentations at a later retention time, indicating the presence of another diastereomer(s) that is presumably the (*Z*,*Z*)- or (*Z*,*E*)-isomers. The spiroacetal 2-methyl-8-propyl-1,7-dioxaspiro[5.5]undecane (**14**) was assigned by examining a molecular ion of *m*/*z* 212 and the fragment ions at *m*/*z* 143 and 140, indicative of propyl substituted, and at *m*/*z* 115, 112, indicative of methyl substituted six-membered rings. These patterns match those in the previous study [20]. Another dihydropyran 6-hexyl-2-methyl-3,4-dihydro-2*H*-pyran (**13**) exhibited similar fragmentation patterns to 2-methyl-6-pentyl-3,4-dihydro-2*H*-pyran (**8**) (Table 1), but molecular ion was observed at *m*/*z* 182, indicating a very similar structure with one carbon higher. These newly reported compounds occur at trace-level quantities, and this is probably why they were not detected previously [19]. The present study was able to detect these additional compounds, of which 2,8-dimethyl-1,7-dioxaspiro[5.5]undecane isomer (**9**) and 2-ethyl-7-methyl-1,6-dioxaspiro[4.5]decane isomer 2 (**11**) elicited very strong EPD signals and 6-hexyl-2-methyl-3,4-dihydro-2*H*-pyran (**13**) elicited EAD signals (Figure 2). These EAD and EPD responses prompted closer examination of the trace-level compounds in the present study. Interestingly, the amide *N*-(3-methylbutyl)acetamide (**4**) was a major compound in the male headspace samples but was minor in the rectal glands. Retention times and, hence, retention indexes of *N*-(3-methylbutyl)acetamide (**4**) and (*E*,*E*)-2,8-dimethyl-1,7-dioxaspiro[5.5]undecane (**5**) are similar (see RIs in Table 1), suggesting similarities in their boiling points and solubility in the stationary phase. Indeed, the boiling points of *N*-(3-methylbutyl)acetamide (**4**) and (*E*,*E*)-2,8-dimethyl-1,7-dioxaspiro[5.5]undecane (**5**) are 220–224 °C (Dehn 1912) and 225 °C [31], respectively, which indicates they would have similar volatilities. Thus, large differences in the rate of condensation are not expected. It is possible that other secretory glands also produce and release *N*-(3-methylbutyl)acetamide (**4**), contributing to a larger proportion in the air. For 2-hydroxyundecan-6-one, in *B. tryoni* and *Z. cucumis*, intermediates in the biosynthesis of (*E*,*E*)-2,8-dimethyl-1,7-dioxaspiro[5.5]undecane (**5**) originate from a fatty acid via different pathways, but 2-hydroxyundecan-6-one is the common and advanced intermediate [32]. It is likely that *B. kraussi* also employs another pathway to generate a unique molecular signature. Interconversion between 2-hydroxyundecan-6-one and 2-methyl-6-pentyl-3,4-dihydro-2*H*-pyran (**8**) observed is facile and occurs via a tetrahydropyranol, an immediate precursor of (*E*,*E*)-2,8-dimethyl-1,7-dioxaspiro[5.5]undecane (**5**) [32,33]. The absence of the compound, therefore, may be related to the interconversion process, where the compound might not have been present during the time of the day when samples were collected.

Sexually mature female *B. kraussi* released a more complex blend of lower volatility compounds than males. The major compounds ethyl dodecanoate (**19**) and ethyl tetradecanoate (**25**) in the present study have also been reported as the major constituents of the rectal glands and headspace volatiles of females in several other *Bactrocera* [8,22,23,26,34]. The esters of saturated/unsaturated fatty acids and several acids are female specific and have also been reported in other *Bactrocera* species but in different proportions in each species [8,22,23,24,26,34,35,36]. Interestingly, ethyl dodecanoate (**19**) comprised 39.1% in rectal gland samples but 70.2% in headspace samples. Ethyl dodecanoate (**19**) is more volatile than the homologous ester series with higher carbon numbers [32,37,38]. The higher volatility may have resulted in the higher proportion in the air than the ethyl esters of higher carbons, such as ethyl tetradecanoate (**25**) or ethyl hexadecanoate (**33**). EAD detected antennal responses of males to ethyl dodecanoate (**19**), ethyl tetradecanoate (**25**) and ethyl (*E*)-hexadec-9-enoate (**31**), indicating a possible pheromone role of these female-specific compounds. In *B. oleae*, methyl dodecanoate (**17**), ethyl dodecanoate (**19**), ethyl tetradecanoate (**25**) and ethyl hexadecanoate (**33**) are EAD active and ethyl dodecanoate (**19**) and methyl hexadecanoate (**28**) attract conspecific females and males, respectively [35]. In *B. musae*, (*E*,*E*)-2,8-dimethyl-1,7-dioxaspiro[5.5]undecane (**5**) and ethyl dodecanoate (**19**) are EAD active [22].

Strong EPD responses to the four spiroacetals indicate that palps are more sensitive to the endogenous compounds than antennae in *B. kraussi*. Palps of both sexes responded strongly to 2,8-dimethyl-1,7-dioxaspiro[5.5]undecane isomer (**9**) and 2-ethyl-7-methyl-1,6-dioxaspiro[4.5]decane isomer (**11**) that were present only in very minute amounts. These EPD results emphasize a high sensitivity of the EAD/EPD system, as well as the likely importance of the minor compounds in chemical communication. These compounds are volatile and these findings suggest that *B. kraussi* may utilize palps to detect some long-range chemical signals. Differences in olfactory function between antennae and maxillary palps are known in other *Bactrocera* species, including *B. tryoni* [37,38] and *B. depressa* [39]. EPD responses of males of these species to cuelure, a male lure, were higher than EAD responses, suggesting that palps of those species might serve olfactory functions for some long-range odorants [37,39]. The major spiroacetal (*E*,*E*)-2,8-dimethyl-1,7-dioxaspiro[5.5]undecane (**5**) may be of particular biological importance as it elicited antennal and palpal responses in both sexes.

## 4. Materials and Methods

### 4.1. Insects

A laboratory-reared population of *B. kraussi* (G27), maintained using a carrot-based larval diet, was obtained from the Queensland Department of Agriculture and Fisheries (Cairns). Approximately 500 pupae were placed in a 47.5 × 47.5 × 47.5 cm fine mesh cage (Megaview Bugdorm 4S4545, Taiwan) for emergence and kept in a controlled environment room at 25 ± 0.5 °C, 65 ± 5% relative humidity (RH) and 11.5:0.5:11.5:0.5 light/dusk/dark/dawn photoperiod at Macquarie University, Sydney (Australia). Adult flies were fed with sugar and yeast hydrolysate (MP Biomedicals LLC) provided separately and tap water through a soaked sponge. Flies were reared for one generation at Macquarie University using a standard carrot diet [40] and following the previous methodology [41]. Flies were separated by sex within 3 days after emergence and transferred to 12.5 L clear plastic cages (about 180 flies per cage). No mating was observed before separating the flies by sex. All flies were kept with the same diet and environmental conditions described above. All experiments used 12- to 18-day old virgin flies.

### 4.2. Rectal Gland Extraction

Rectal glands were excised from sexually mature males and females, prepared as described in Section 4.1. Handling of the flies and the gland extractions followed a known procedure [18]. In brief, flies in a plastic vial (5 mL) were killed by placing them on dry ice. The abdomen was gently squeezed with tweezers such that the glands protruded slightly. The glands were then gently pulled out with tweezers, and the secretory sac was separated. Glands were carefully placed in a pre-cooled tear-drop vial in dry ice. Once 20 glands were collected, the vials were removed from the dry ice and transferred to a rack. An aliquot of 200 μL of *n*-hexane (HPLC grade, Sigma-Aldrich, St. Louis, MO, USA) was added to the vials and the vials were allowed to stand at room temperature for 10 min. The extracts were then transferred to a new vial and stored at −20 °C until analysed. Six replicates per sex were collected using 20 glands per replicate.

### 4.3. Headspace Collection

Headspace samples were collected through the immediate pre-dusk (30 min), dusk (30 min) and immediate post-dusk (30 min) light phases, a total of 1.5 h adsorption, in a controlled environment room under the same conditions under which the flies were kept (25 ± 0.5 °C, 65 ± 5% RH). The experiment time of the day was selected based on our observations that male calling and subsequent mating occur at dusk. Thirty sexually mature males and 30 sexually mature females of *B. kraussi* were separately placed into glass chambers (150 mm long × 40 mm ID) 30 min before dusk and charcoal-filtered air was drawn over the flies at a flow rate of 1.0 L/min (air pulling system). Volatile compounds in headspace were adsorbed onto 50 mg of Tenax adsorbent (Scientific Instrument Services, Inc., Ringoes, NJ, USA, Tenax-GR Mesh 60/80) packed into glass cartridges (6 mm OD × 50 mm length) and fitted with deactivated glass wool plugs. To distinguish any contaminants, a blank sample, comprising an empty glass chamber, was run and analysed along with each volatile collection. Adsorbed volatiles in each Tenax were subsequently eluted with 1 mL of *n-*hexane (HPLC grade, Sigma-Aldrich) into a standard 1.5 mL GC vial. Sample vials were stored at −20 °C until analysed. Five replicates per sex were collected. Tenax traps were conditioned at 200 °C for three hours under a nitrogen stream (75 mL/min) prior to each experiment. Glass chambers were washed with 5% Extran aqueous solution, rinsed with hot tap water, and conditioned at 200 °C for 18 h. Activated charcoal filters were thermally conditioned by heating them at 200 °C for 18 h prior to each experiment [42].

### 4.4. Analysis of Rectal Gland Extracts and Headspace Collections

Mass spectra were recorded by gas chromatography–mass spectrometry (GC–MS) on a Shimadzu GCMS-QP2010 or GCMS-QP2020 NX (Shimadzu Corporation, Nakagyo-ku, Kyoto, Japan) equipped with an AOC-20i or AOC-6000 autosampler, respectively, and a split/splitless injector, a SH-Rtx-5Sil MS or Rtx-35 fused silica capillary column (30 m × 0.25 mm, 0.25 μm film) and integrated mass spectrometer (MS). Helium (99.999%) (ultra-high purity, BOC, Australia) was a carrier gas with a constant flow of 1.5 mL/min. The injector and MS transfer line temperature were both set to 250 °C. The temperature program for rectal glands was set at 50 °C (1 min) and increased to 260 °C (1 min) at a rate of 15 °C/min. The temperature program for headspace was set at 50 °C (4 min), and increased to 250 °C (6 min) at a rate of 10 °C/min. The ion source (EI 70 eV) temperature was set at 200 °C. The samples were analysed by injecting 1 μL of the sample solution at splitless mode for headspace samples and split mode for gland extracts at a split ratio of 10, with a sampling time of 1 min. Impurities were identified through comparison with the air samples or solvent runs. Chambers were washed with 5% Extran aqueous solution, rinsed with hot tap water, and conditioned at 200 °C for 18 h.

Compounds including esters, amides and some spiroacetals were identified through comparison with gas chromatography retention times and mass spectra of authentic samples. Of the 36 compounds detected in *B. kraussi*, 18 were commercially available and were purchased from Sigma-Aldrich (Saint Louis, MO, USA), Alfa-Aesar (Haverhill, MA, USA), and Nu-Chek-Prep, INC (Minneapolis, MN, USA). This included 2-ethyl-1-hexanol (≥98%) (**1**), diethyl butanedioate (99%) (**7**), methyl dodecanoate (≥98%) (**17**), dodecanoic acid (≥98%) (**18**), ethyl dodecanoate (≥98%) (**19**), ethyl tridecanoate (99%) (**20**), methyl tetradecanoate (≥98%) (**22**), tetradecanoic acid (≥98%) (**23**), ethyl (*Z*)-tetradec-9-enoate (97%) (**24**), ethyl tetradecanoate (99%) (**25**), 3-methylbutyl dodecanoate (≥97%) (**26**), methyl (*Z*)-hexadec-9-enoate (≥99%) (**27**), methyl hexadecanoate (≥99%) (**28**), (*Z*)-hexadec-9-enoic acid (≥98.5%) (**29**), hexadecanoic acid (≥99%) (**30**), ethyl hexadecanoate (≥99%) (**33**), (*Z*)-octadec-9-enoic acid (≥99%) (**34**), and ethyl (*Z*)-octadec-9-enoate (98%) (**36**). (*E*,*E*)-2,8-Dimethyl-1,7-dioxaspiro[5.5]undecane (**5**), 2-ethyl-7-methyl-1,6-dioxaspiro[4.5]decane (**6**), *N*-(2-methylbutyl)acetamide (**3**), *N*-(3-methylbutyl)acetamide (**4**), 6-oxononan-1-ol (**16**), propyl dodecanoate (**21**), ethyl (*Z*)-hexadec-9-enoate (**31**) and ethyl (*E*)-octadec-9-enoate (**35**) were not available commercially, and were synthesised following literature procedures (see Section 4.5). 2-Methyl-1,6-dioxaspiro[4.5]decane (**2**), 2-methyl-6-pentyl-3,4-dihydro-2*H*-pyran (**8**), (*E*,*E*)-2-ethyl-8-methyl-1,7-dioxaspiro[5.5]undecane (**10**), 2-ethyl-7-methyl-1,6-dioxaspiro[5.5]undecane isomer 2 (**11**), 2,8-dimethyl-1,7-dioxaspiro[5.5]undecane isomers (**9** and **12**), 6-hexyl-2-methyl-3,4-dihydro-2*H*-pyran (**13**), 2-methyl-8-propyl-1,7-dioxaspiro[5.5]undecane (**14**), 2,8-dimethyl-1,7-dioxaspiro[5.5]undecan-3-ol (**15**) and ethyl (*E*)-hexadec-9-enoate (**32**) were tentatively identified based on the literature mass spectral fragmentation patterns [4,19,20]. The relative proportion of a compound in samples of rectal glands or headspace may provide an important clue with which potential applications can be studied. Hence, the relative percentage of each compound in the rectal gland blend or headspace was obtained by dividing an individual peak area by the total peak area and multiplying the result by 100.

### 4.5. Synthesis

#### 4.5.1. General Procedure

All reagents were purchased from Sigma-Aldrich and used without further purification. All solvents were anhydrous or analytical grade (Sigma-Aldrich) and used without further purification. The reaction progress was monitored by GC–MS (using the procedure given in Section 4.4). Solvents were removed under reduced pressure using a Büchi Rotavapor R-200 and Büchi B-490 heating bath set to 40 °C. Mixtures were further dried under high vacuum using an Alcatel Pascal 2005SD vacuum pump. NMR spectra were recorded on a Bruker AVANCE-400 instrument (^1^H NMR: 400 MHz, ^13^C NMR: 101 MHz) or a Bruker AVANCE-600 instrument equipped with a cryoprobe (^1^H NMR: 600 MHz, ^13^C NMR: 150 MHz) using CDCl_3_ and C_6_D_6_. The ^1^H NMR chemical shifts were referenced to the residual protonated solvent peaks at δH 7.26 for chloroform-d and 7.15 for benzene-*d*_6_. ^13^C NMR chemical shifts were referenced to the central solvent peaks of bulk solvent at δC 77.16 for chloroform-d and 127.68 for benzene-*d*_6_. J values are given in Hz.

#### 4.5.2. Synthesis of *N*-(2-Methylbutyl)acetamide (**3**)

The synthesis was conducted using the previously described method (Scheme 1) [43]. Acetic anhydride (7.7 g, 75 mmol) was added to a mixture of 2-methylbutylamine (4.4 g, 50 mmol) in water (50 mL). The clear reaction mixture was stirred at room temperature for 0.5 hour and the completion of the reaction at this time was determined by GC–MS. The clear reaction mixture was extracted with ethyl acetate (3 × 50 mL). The combined organic layer was washed with 5% aqueous sodium bicarbonate solution (150 mL), dried over sodium sulfate and concentrated under reduced pressure to give the crude product, which was purified by vacuum distillation (150*–*160 °C, 20 mm Hg) to afford *N*-(2-methylbutyl)acetamide (**3**) as a clear liquid (5.6 g, 86%). ^1^H NMR (400 MHz, CDCl_3_) δ 0.83 (6H, m, CHCH_2_C**H**_3_ and CH_2_C**H**_3_), 1.07 (1H, apparent sep, *J* = 6.6, C**H**CH_3_), 1.29–1.53 (2H, m, C**H**_2_CH_3_), 1.93 (3H, s, C**H**_3_CO), 2.94–3.13 (2H, m, NC**H**_2_), 6.33 (1H, bs, N**H**). ^13^C NMR (101 MHz, CDCl_3_) δ 11.2, 17.1, 23.1, 27.0, 34.8, 45.4, 170.6. GC–MS (EI) *m*/*z* (%) 129 (M^+^, 8), 100 (M^+^−CH_2_CH_3_, 38), 72 (M^+^−CHCH(CH_3_)CH_2_CH_3_, 100). Spectral data were consistent with the literature [41].

#### 4.5.3. Synthesis of *N*-(3-Methylbutyl)acetamide (**4**)

Using a similar reaction, work-up and purification conditions to *N*-(2-methylbutyl)acetamide (**3**) (Scheme 2), 3-methylbutylamine (5.0 g, 57 mmol) in water (50 mL) was acetylated with acetic anhydride (8.7 g, 86 mmol) to produce *N*-(3-methylbutyl)acetamide (**4**) as a clear liquid (5.4 g, 73%). ^1^H NMR (400 MHz, CDCl_3_) δ 0.85 (6H, d, *J* = 6.6, CH(C**H**_3_)_2_), 1.33 (2H, m, C**H**_2_CH_3_), 1.56 (1H, sep, *J* = 6.7, C**H**), 1.92 (3H, s, C**H**_3_CO), 3.18 (2H, m, NC**H**_2_), 6.21 (1H, bs, N**H**). ^13^C NMR (101 MHz, CDCl_3_) δ 22.4, 23.1, 25.8, 38.0, 38.3, 170.4. GC–MS (EI) *m*/*z* (%) 129 (M^+^, 5), 114 (M^+^−CH_3_, 12), 73 (M^+^−CH_2_CH_2_CH(CH_3_)_2_, 100). Spectral data were consistent with the literature [41].

#### 4.5.4. Synthesis of 2,8-Dimethyl-1,7-dioxaspiro[5.5]undecane

1,10-Undecadien-6-ol. Following the previously described metho (Scheme 3) [18], Grignard reaction followed by hydrolysis was conducted to give 1,10-undecadien-6-ol. In brief, a flame-dried argon-flushed two-necked round bottom flask was charged with magnesium (1.8 g, 74 mmol), a single crystal of iodine, and a magnetic stirrer bar, and fitted with a condenser. Dry diethyl ether (60 mL) was added and the suspension was brought to reflux. 5-Bromopent-1-ene (10 g, 67 mmol) in diethyl ether (30 mL) was added dropwise and then the colourless suspension was stirred at reflux for 4 h. The colourless suspension was cooled to 0 °C and ethyl formate (2.6 g, 34 mmol) was added. The suspension was warmed to room temperature, stirred for 1 hour, then quenched with saturated ammonium chloride and extracted with diethyl ether (3 × 15 mL). The combined organic layers were washed with saturated aqueous brine and dried over magnesium sulfate. After solvent removal by rotary evaporation, the yellow oil was refluxed in 15% aqueous potassium hydroxide solution for 3 h. The solution was cooled to room temperature and extracted with diethyl ether (3 × 20 mL). Solvent was removed under reduced pressure to give the crude product as a yellow oil, which was purified by distillation (110–115 °C, 10 mm Hg) to afford 1,10-undecadien-6-ol as a colourless oil (3.7 g, 60% yield). ^1^H NMR (400 MHz, CDCl_3_) δ 5.81 (2H, ddt, *J* = 17, 10.3, 6.7 Hz, C**H**=), 5.01 (2H, dq, *J* = 17.1, 1.7 Hz, C**H_2_**=), 4.91–5.01 (2H, m, C**H_2_**=), 3.61 (1H, bs, C**H**OH), 2.00–2.13 (4H, m, C**H**_2_CH=CH_2_), 1.26–1.61 (9H, m, including OH). ^13^C NMR (101 MHz, CDCl_3_) δ 138.7 (CH=), 114.6 (CH**_2_**=), 71.7 (CHOH), 36.9 (CH_2_), 33.7 (CH_2_), 24.9 (CH_2_). GC–MS (EI) *m*/*z* (%) 84 (12.3), 81 (100), 80 (10), 79 (19.5), 69 (9.2), 68 (9.3), 67 (20.6), 58 (9.5), 57 (30.2), 55 (72.3), 54 (27.7), 53 (9.4), 43 (32.1), 42 (9), 41 (43.8). Spectral data were consistent with the literature [18].

Undeca-1,10-dien-6-one. Freshly prepared Jones reagent (2.7 g of chromium trioxide in 4 mL of sulfuric acid and 12 mL of distilled water) was added dropwise to a solution of 1,10-undecadien-6-ol (3.99 g, 23.7 mmol) in acetone (10 mL) at −10 °C, and the reaction monitored by GC–MS. After completion of the reaction (2 h), the green suspension was filtered through a pad of Celite. The filtrate was washed with saturated aqueous sodium bicarbonate (17 mL), extracted with diethyl ether (4 × 50 mL) and washed with water (50 mL) and saturated aqueous brine (50 mL), then dried over magnesium sulfate. Concentration by rotary evaporation yielded undeca-1,10-dien-6-one as a colourless oil (2.9 g, 75% yield). ^1^H NMR (400 MHz, CDCl_3_) δ 5.80 (2H, ddt, *J* = 17.2, 10.3, 6.7 Hz, C**H**=), 4.87–4.97 (4H, m, C**H_2_**=), 2.33 (4H, t, *J* = 7.5 Hz, C**H**_2_CO), 1.98 (4H, m, C**H**_2_CH=CH_2_), 1.60 (4H, quin, *J* = 7.3 Hz, CH_2_C**H**_2_CH_2_). ^13^C NMR (101 MHz, CDCl_3_) δ 210.9 (CO), 138.0 (CH=), 115.2 (CH**_2_**=), 41.9 (CH_2_), 33.1 (CH_2_), 22.8 (CH_2_). GC–MS (EI) *m*/*z* (%) 112 (14.6), 97 (30), 84 (27.8), 83 (10.6), 70 (14.1), 69 (59.5), 58 (48.7), 55 (49.6), 43 (24.5), 41 (100). Spectral data were consistent with the literature [18].

2,8-Dimethyl-1,7-dioxaspiro[5.5]undecane. Hg(OAc)_2_ (1.9 g, 6.1 mmol) was added to a stirred solution of undeca-1,10-dien-6-one (0.5 g, 3 mmol) in 1% aqueous perchloric acid: tetrahydrofuran (15 mL:15 mL) and the solution was stirred for 15 h. Benzyltriethylammonium chloride (2.4 g, 10.5 mmol) in 10% aqueous sodium hydroxide (15 mL) and dichloromethane (5 mL) was added followed by sodium borohydride (0.09 g, 2.3 mmol) in 10% aqueous sodium hydroxide (5 mL). The gray suspension was stirred and monitored by GC–MS. After completion of the reaction (20 min), the gray suspension was filtered through a pad of Celite, which was then washed with 30 mL of diethyl ether. The aqueous phase was then extracted with diethyl ether (3 × 30 mL) and the combined organic layer (from Celite wash and extraction) were washed with saturated aqueous brine (50 mL) and dried over magnesium sulfate. After solvent removal by rotary evaporation the product was purified by Kugelrohr distillation (bp 110 °C; 30 mm Hg). According to the literature [18] under this condition a mixture of (*E*,*E*)-diastereomer with some (*E*,*Z*) and no (*Z*,*Z*) isomer is obtained. These conformational isomers produced different MS fragmentation patterns that were matched with those in the literature [18].

(*E*,*E*)-2,8-Dimethyl-1,7-dioxaspiro[5.5]undecane (**5**) [44]. ^13^C NMR (101 MHz, C_6_D_6_) δ 95.75 (CO), 64.8 (CO), 35.33 (CH**_2_**), 32.90 (CH_2_), 21.92 (CH_3_), 19.03 (CH_2_). GC–MS (EI) *m*/*z* (%) 184 (M^+^, 5.6), 169 (M^+^−CH_3_, 1.9), 140 (M^+^−CH_3_CHO, 11.6), 125 (8.2), 115 (CH_3_(C_5_H_7_O)=OH^+^, 92.4), 114 (43.2), 113 (8.6), 112 (CH_3_(C_5_H_7_O)=CH_2_, 100), 97 (68.4), 84 (15.7), 83 (23.8), 73 (24.8), 71 (18.5), 70 (15.8), 69 (54.9), 58 (18.2), 55 (52.1), 43 (69.4), 42 (35.9), 41 (56.1).

(*E*,*Z*)-2,8-Dimethyl-1,7-dioxaspiro[5.5]undecane (**9**). GC–MS (EI) *m*/*z* (%) 184 (M^+^, 8.1), 115 (CH_3_(C_5_H_7_O)=OH^+^, 100), 114 (37), 112 (CH_3_(C_5_H_7_O)=CH_2_, 39.5), 97 (73.1), 83 (11.8), 73 (27.5), 71 (12.4), 69 (59.9), 55 (41.8), 43 (38.6), 42 (24.6), 41 (38.4).

#### 4.5.5. Synthesis of 2-Ethyl-7-methyl-1,6-dioxaspiro[4.5]undecane (**6**)

(*E*)-Ethyl hex-4-enoate. Following the previously described method (Scheme 4) [45], ortho-ester Johnson-Claisen rearrangement was conducted to give (*E*)-ethyl hex-4-enoate. In brief, 3-Buten-2-ol (6 g, 84.3 mmol), triethylorthoacetate (20.3 g, 125.4 mmol) and acetic acid (0.1 g, 2.5 mmol) was heated at 140 °C for 4 h. The consumption of starting material at this time was determined by GC–MS. The reaction mixture was then cooled to room temperature and ethanol (20 mL) and water (20 mL) was added. The aqueous layer was extracted with diethyl ether (3 × 10 mL). The combined organic layers were stirred with hydrochroric acid solution (1 M aq, 20 mL) at room temperature for 30 min, then washed with brine (20 mL) and dried over magnesium sulfate. Concentration by rotary evaporation yielded (*E*)-ethyl hex-4-enoate as a clear yellow oil (10.3 g, 86%). GC–MS (EI) *m*/*z* (%) 142 (6.6), 97 (26.3), 96 (14.9), 88 (22.3), 71 (58.1), 70 (15.6), 69 (83.0), 68 (100), 67 (29.8), 60 (32.6), 55 (74.6), 43 (19.3), 42 (17.3), 41 (82.55). Experimental spectra were consistent with literature data [46].

(*E*)-Hex-4-enoic acid. Minor modification was made to the method of Tay et al. (2016) to form (*E*)-hex-4-enoic acid [45]. A solution of sodium hydroxide in 1:1 water:methanol (3.6 M, 100 mL) was added to a solution of (*E*)-ethyl hex-4-enoate (10.31 g, 72.5 mmol) in tetrahydrofuran (50 mL). The reaction mixture was stirred at 60 °C. The consumption of starting material at this time was determined by GC–MS. The reaction mixture was then cooled to room temperature and diethyl ether (40 mL) and sodium hydroxide (1 M aq, 40 mL) was added. The aqueous layer was extracted with diethyl ether (3 × 40 mL) and the combined organic layers were washed with sodium hydroxide (1 M aq, 3 × 20 mL). The combined aqueous washes were acidified to pH = 1 using hyrdochoric acid (1 M aq), extracted with diethyl ether (3 × 40 mL), then combined organic layers were washed with brine, dried over magnesium sulfate and the solvent removed by rotary evaporation yielded (*E*)-hex-4-enoic acid as a clear oil which was used in the next step without further purification (7.5 g, 91%). ^1^H NMR (400 MHz, CDCl_3_) δ 5.50–5.42 (2H, m, CH_3_C**H**C**H** CH_2_), 2.44–2.40 (2H, m, C**H**_2_), 2.35–2.29 (2H, m, CH_2_), 1.67–1.65 (3H, m, CH_3_). Experimental spectra were consistent with literature data [47].

(*E*)-N-Methoxy-N-methylhex-4-enamide. Distilled trimethylamine (10 g, 100 mmol) was added to a solution of (*E*)-hex-4-enoic acid (4 g, 35 mmol) in dichloromethane (100 mL) at 0 °C, followed by the addition of *N*,*O*-dimethylhydroxylamine hydrochloride (3.4 g, 35 mmol) in one portion. After stirring for 10 min, *N*-(3-dimethylaminopropyl)-*N*′-ethylcarbodiimide hydrochloride (6.7 g, 35 mmol) was added in two equal portion over 5 min and the heterogeneous mixture was warmed to room temperature. After stirring for 12 h, water (100 mL) was added and the aqueous layer was extracted with dichloromethane (3 × 30 mL). the combined organic layers were washed with hydrochloric acid (1 M aq, 200 mL), sodium bicarbonate (100 mL), brine (50 mL) and dried over magnesium sulfate. After solvent removal by rotary evaporation the product was purified by Kugelrohr distillation (bp 95–105 °C; 15 mm Hg) to yield (*E*)-*N*-methoxy-*N*-methylhex-4-enamide as a clear oil (2.45 g, 45%). ^1^H NMR (400 MHz, CDCl_3_) δ 5.50–5.47 (2H, m, CH_3_C**H**C**H** CH_2_), 3.69 (3H, s, OC**H**_3_), 3.19 (3H, s, NC**H**_3_), 2.51–2.47 (2H, m, C**H**_2_), 2.34–2.29 (2H, m, C**H**_2_), 1.65 (3H, d, *J* = 6.6, C**H**_3_). Experimental spectra were consistent with literature data [47].

(*E*)-Undeca-1,9-dien-6-one. To a pre-cooled (−10 °C) Grignard regent prepared from 5-bromo-1-pentene(0.6 g, 4 mmol), magnesium (1.0 g, 4.4 mmol) and a single crystal of iodine in diethyl ether (4 mL) was added (*E*)-undeca-1,9-dien-6-one (0.6 g, 4 mmol) in diethyl ether (8 mL) over 10 min, then slowly warmed to room temperature and stirred for 24 h. Then diethyl ether (20 mL) and saturated ammonium chloride (20 mL) was added and the aqueous layer was extracted with diethyl ether (2 × 10 mL). The combined organic layers were washed with brine (10 mL) and dried over magnesium sulfate. Solvent removal by rotary evaporation yielded a crude yellow oil which then purified by flash column chromatography (0:100–10:90 ethyl acetate:hexanes) to yield (*E*)-undeca-1,9-dien-6-one as a clear oil (0.2 g, 30%). ^1^H NMR (400 MHz, CDCl_3_) δ 5.74–5.64 (1H, m, CH_2_C**H**CH_2_), 5.40–5.29 (2H, m, C**H**C**H**), 4.96–4.88 (2H, m, CHC**H**_2_), 2.39–2.31 (4H, m, C**H**_2_COC**H**_2_), 2.20–2.15 (2 H, m, CHC**H**_2_CH_2_), 2.00–1.95 (2H, m, CHC**H**_2_CH_2_), 1.60 (5H, m, CH_2_C**H**_2_CH_2_ and C**H**_3_). ^13^C NMR (101 MHz, CDCl_3_) δ 210.5 (C=O), 138.0 (CH), 129.6 (CH), 125.8 (CH), 115.1 (CH_2_), 42.6 (CH_2_), 42.0 (CH_2_), 33.1 (CH_2_), 26.8 (CH_2_), 22.7 (CH_2_), 17.8 (CH_3_). This compound is known, but spectral data are not available in the literature.

2-Ethyl-7-methyl-1,6-dioxaspiro[4.5]undecane. Using a similar reaction and work-up conditions to 2,8-dimethyl-1,7-dioxaspiro[5.5]undecane, oxymercuration-reduction was performed to convert (*E*)-undeca-1,9-dien-6-one (0.1 g, 6 mmol) to 2-ethyl-7-methyl-1,6-dioxaspiro[4.5]undecane. Under this condition, a mixture of (*E*,*E*)- and (*E*,*Z*)-isomers of 2-ethyl-7-methyl-1,6-dioxaspiro[4.5]undecane was formed together with (*E*,*E*)- and (*E*,*Z*)- isomers of 2,8-dimethyl-1,7-dioxaspiro[5.5]undecane (clear oil, 0.4 g, 36%).

(*E*,*E*)-2-Ethyl-7-methyl-1,6-dioxaspiro[4.5]undecane (**5**). GC–MS (EI) *m*/*z* (%) 184 (M^+^, 0.9), 169 (M^+^−CH_3_, 0.5), 155 (M^+^−CH_2_CH_3_, 7.6), 140 (M^+^−CH_3_CHO, 2.47), 126 (2.3), 115 (CH_3_(C_5_H_7_O)=OH^+^, 41.13), 114 (11.9), 113 (4.13), 112 (M^+^−CH_2_CHCH(OH)CH_3_, 30.7), 97 (51.82), 95 (10.2), 85 (51.7), 83 (23.4), 73 (22.9), 71 (14.5), 70 (14.7), 69 (64.3), 67 (9.7), 55 (100), 58 (16,2), 43 (89.2), 42 (44.6), 41 (79.3). Experimental spectra were consistent with literature data [33].

#### 4.5.6. Synthesis of 6-Oxanon-1-ol (**11**)

The synthesis was conducted using the previously described method (Scheme 5) [48]. A mixture of ethanol-water (9:1, 24 mL), zinc dust (1.0 g, 16 mmol), copper(I) iodide (0.9 g, 4.8 mmol), 3-bromo-1-propanol (880 mg, 6.4 mmol) in ethanol (4 mL) and 1-hexen-3-one (620 mg, 6.4 mmol) in ethanol (4 mL) at 0 °C was sonicated for 7.5 h, and reaction progress was monitored by GC–MS. The completion of the reaction at this time was determined by GC–MS. The reaction was then quenched with brine and filtered. Concentration by rotary evaporation yielded the crude product that was dissolved in diethyl ether (50 mL), washed with water (2 × 20 mL) and brine (2 × 20 mL), and dried over Na_2_SO_4_. Solvent was removed under reduced pressure, yielding the crude product, which was purified by flash column chromatography (eluted twice with 0–10% ethyl acetate in n-hexane) to afford 6-oxanon-1-ol as a colourless oil (112 mg, 11%). GC–MS (EI) *m*/*z* (% of base peak) 158 (M^+^, 1.1), 140 (M^+^−H_2_O, 2.1), 115 (8.2), 112 (3.5), 99 (7.3), 97 (26.1), 86 (32.0), 79 (10.0), 73 (11.4), 71 (66.9), 69 (70.1), 58 (52.1), 55 (34.7), 43 (100), 41 (72.5). Experimental spectra were consistent with literature data [49].

#### 4.5.7. Synthesis of Propyl Dodecanoiate (**15**)

A mixture of dodecanoic acid (1.0 g, 5 mmol), 1-propanol (10 mL) and concentrated sulfuric acid (2 drops) was heated to reflux for 1.5 h. After cooling, diethyl ether (10 mL) and 5% *w*/*v* aqueous sodium bicarbonate (10 mL) were added to the reaction mixture. The organic layer was separated and washed with 5% *w*/*v* aqueous sodium bicarbonate (3 × 10 mL) and dried over sodium sulfate (Scheme 6). The solvent was removed under reduced pressure, yielding the crude product, which was purified by distillation to give propyl dodecanoate as a colourless liquid (0.44 g, 38% yield). ^1^H NMR (600 MHz, CDCl_3_) δ 5.32–5.35 (2H, m, C**H**=C**H**), 4.11 (2H,4.02 (2H, t, *J* = 6.7 Hz, C**H**_2_OCO), 2.29 (2H, t, *J* = 7.5 Hz, C**H**_2_COOPr), 1.57–1.66 (4H, m, C**H**_2_CH_2_COOPr, CH_3_C**H**_2_CH_2_OCO), 1.25–1.29 (16H, m, C**H**_2_), 0.93 (3H, t, *J* = 7.4 Hz, OCH_2_CH_2_C**H**_3_), 0.87 (3H, t, *J* = 7.0 Hz, CH_2_C**H**_3_). ^13^C NMR (150 MHz, CDCl_3_) δ 174.1 (C=O), 65.9 (OCH_2_), 34.5 (CH_2_), 32.0 (CH_2_), 29.7 (CH_2_), 29.6 (CH_2_), 29.47 (CH_2_), 29.40 (CH_2_), 29.3 (CH_2_), 25.1 (CH_2_), 22.8 (CH_2_), 22.1 (CH_2_), 14.2 (CH_3_), 10.5 (CH_3_). GC–MS (EI) *m*/*z* (%) 242 (M^+^, 4.3), 213 (1.4), 201 (M^+^−CH_2_CH_2_CH_3_, 27.5), 183 (M^+^−OCH_2_CH_2_CH_3_, 25.7), 171 (6.6), 157 (6.8), 143 (3.3), 129 (8.7), 115 (21.8), 102 (32.5), 97 (7.7), 85 (12.2), 73 (39.3), 61 (100), 57 (30.9), 43 (80.2).

#### 4.5.8. Synthesis of Ethyl (*Z*)-Hexadec-9-enoate (**31**)

Using similar conditions used in the synthesis of propyl dodecanoate (Scheme 7), (*Z*)-hexadec-9-enoic acid (0.50 g, 1.9 mmol), was esterified with ethanol (10 mL) in the presence of concentrated sulfuric acid (2 drops), quenched with sodium bicarbonate and diethyl ether and purified by distillation to afford ethyl (*Z*)-hexadec-9-enoate as a colourless oil (0.11 g, 19%). ^1^H NMR (400 MHz, CDCl_3_) δ 5.32–5.35 (2H, m, C**H**=C**H**), 4.11 (2H, q, *J* = 7.2 Hz, OC**H_2_**CH_3_), 2.28 (2H, t, *J* = 7.5 Hz, C**H_2_**COOEt), 1.98–2.01 (4H, m, C**H_2_**CH=CHC**H_2_**), 1.59–1.63 (2H, m, C**H_2_**CH_2_COOEt), 1.23–1.30 (19H, m, C**H_2_**), 0.88 (3H, t, *J* = 6.9 Hz, CH_2_C**H_3_**). ^13^C NMR (101 MHz, CDCl_3_) δ 174.0 (C=O), 130.1 (CH), 129.9 (CH), 60.2 (OCH_2_), 34.5 (CH_2_), 31.9 (CH_2_), 29.87 (CH_2_), 29.82 (CH_2_), 29.3 (CH_2_), 29.26 (CH_2_), 29.23 (CH_2_), 29.1 (CH_2_), 27.36 (CH_2_), 27.30 (CH_2_), 25.1 (CH_2_), 22.8 (CH_2_), 14.4 (CH_3_), 14.2 (CH_3_). GC–MS (EI) *m*/*z* (%) 282 (M^+^, 3.8), 236 (M^+^−OCH_2_CH_3_, 14.3), 218 (1.4), 207 (1.4), 194 (M^+^−CH_2_COOCH_2_CH_3_, 15.0), 179 (1.65), 165 (2.8), 152 (M^+^−(CH_2_)_4_COOCH_2_CH_3_, 14.9), 138 (6.9), 123 (11.9), 101 (31.8), 88 (50.6), 83 (44.9), 69 (64.1), 55 (100), 41 (81.8). Spectral data were not available in the literature.

#### 4.5.9. Synthesis of Ethyl (*E*)-Octadec-9-enoate (**34**)

Using similar conditions used in the synthesis of propyl dodecanoate (Scheme 8), (*E*)-octadec-9-enoic acid (0.45 g, 1.6 mmol), was esterified with ethanol (10 mL) in the presence of concentrated sulfuric acid (2 drops), quenched with sodium bicarbonate and diethyl ether and purified by distillation to afford ethyl (*E*)-octadec-9-enoate as a colourless oil (113 mg, 23%). ^1^H NMR (400 MHz, CDCl_3_) δ 5.36–5.28 (2H, m, C**H**=C**H**), 4.11 (2H, q, *J* = 7.1 Hz, OC**H_2_**CH_3_), 2.27 (2H, t, *J* = 7.6 Hz, C**H_2_**COOEt), 1.95–1.96 (4H, m, C**H_2_**CH=CHC**H_2_**), 1.57–1.60 (2H, m, C**H_2_**CH_2_COOEt), 1.23–1.28 (23H, m, CH_2_), 0.87 (3H, t, *J* = 6.7 Hz, CH_2_CH_2_C**H_3_**). ^13^C NMR (101 MHz, CDCl_3_) δ 174.0 (C=O), 130.6 (CH), 130.4 (CH), 60.3 (OCH_2_), 34.5 (CH_2_), 32.78 (CH_2_), 32.73 (CH_2_), 32.0 (CH_2_), 29.8 (CH_2_), 29.7 (CH_2_), 29.6 (CH_2_), 29.4 (CH_2_), 29.36 (CH_2_), 29.30 (CH_2_), 29.1 (CH_2_), 25.1 (CH_2_), 22.8 (CH_2_), 14.4 (CH_3_), 14.2 (CH_3_). GC–MS (EI) *m*/*z* (%) 310 (M^+^, 3.5), 281 (M^+^−CH_2_CH_3_, 0.25), 264 (M^+^−OCH_2_CH_3_, 16.2), 222 (11.3), 180 (11.2), 155 (7.0), 138 (5.6), 123 (13.5), 111 (20.6), 97 (38.6), 88 (45.6), 83 (49.0), 69 (69.0), 55 (100), 41 (76.4). Spectral data were consistent with the literature [50].

### 4.6. Electrophysiology

Coupled gas chromatography–electroantennogram detection (GC–EAD) or –electropalpogram detection (GC–EPD) was used to record electrophysiological responses. Whole fly heads were excised and secured between two silver wires with capillary electrodes filled with electrode gel (Spectra 360, Parker Laboratories, Inc., Fairfield, NJ, USA). One electrode was placed at the tip of an antenna or palp and the other electrode at the bottom of the head. The antennal preparation was under charcoal filtered and humidified air flow (400 mL/min) controlled by a flow controller (Syntech Stimulus Controller CS-55, Syntech, Hilversum, The Netherlands). The responses of antennae or maxillary palps were amplified/recorded by a data acquisition controller (IDAC-4, Syntech, Hilversum, The Netherlands) and analysed using GCEAD 2014 software version 1.2.5. Separation of the individual compounds in the extracts was achieved on an Agilent 7890B gas chromatography (GC) system equipped with a split/splitless injector, fused silica capillary column SH-Rtx-5Sil MS (30 m × 0.25 mm ID × 0.25 μm film thickness) and flame ionization detector (FID). Hydrogen (99.999% pure, BOC, North Ryde, Australia) was used as a carrier gas with a constant flow of 2.5 mL/min. An aliquot of 1 µL of a sample was injected at splitless mode. The temperature program was set initially at 50 °C (1 min), then increased to 260 °C (1 min) at a rate of 15 °C/min. The injector and detector temperatures were 270 °C and 290 °C, respectively. The effluent of the column was mixed with 30 mL/min make-up nitrogen gas and split with one part going to the internal FID and 1.5 passed to EAD or EPD through a heated transfer line (TC-02, Syntech, Hilversum, The Netherlands), kept at constant temperature of 200 °C. Rectal gland extracts of both sexes were presented to antennae and palps of both sexes. The identities of FID peaks from the GC–EAD system were confirmed by comparing the retention times of GC–MS peaks that operated on the same type of column and under the same GC conditions.

## Data Availability

All data is contained within the article.

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
