# Peer review of "Electrophysiological Responses of Bactrocera kraussi (Hardy) (Tephritidae) to Rectal Gland Secretions and Headspace Volatiles Emitted by Conspecific Males and Females"

_molecules, 2021, doi:10.3390/molecules26165024_

Round 1
Reviewer 1 Report
This manuscript reports the identification of chemical components in rectal glands of Bactrocera kraussi as well as volatiles emitted by both sexes, and the electrophysiological responses of the compounds using GC-EAD and GC-EPD techniques.
Since the data shown in this report are important for understanding the chemical communication of this pest fruit fly species, I judge that this manuscript is worth publishing in Molecules.
Not only the electrophysiological assays but also some behavioral evidence could be provided even preliminarily using those volatile components characterised from the rectal glands as well as headspace.
According to J. Royer (2015), B. kraussi shows affinity both to cuelure and isoeugenol analogs. It is intriguing to know the rectal components when artificially or naturally administrated/incorporated, and the electrophysiological and behavioural responses of B. kraussi to those possible sequestrates.
The chemical identification of volatiles seems well established, mostly confirmed by comparison either with authentic specimens or syntheses. However, some of the aliphatic acid esters with unsaturations can be given at least according to IUPAC nomenclature system, instead of their common names (unless determined unequivocally via ozonolysis or such), to avoid unnecessary confusions on the position of the double bond and geometry, even if the data apparently matched one of the synthetic analogues - as I was taught in the lepidopteran pheromone chemistry as a caveat. According to the supplementary material section, I see some of those compounds were wrongly illustrated (Figures in page 10) - ethyl palmitoleate is shown as ethyl E9-hexadecenoate (instead of Z9-); and contrarily, ethyl elaidate as the Z-form. Thus, I was confused and also concerned about the identification manner for those isomer-bearing constituents.
M&M 4.2. (L. 233-) Some further descriptions on the sampling methods could help in this case, particularly where the components were detected more in the headspace (e.g. amides) or vice versa. Was the whole rectal glandular organ from females "pulled out" in the same manner as from males?
Reviewer 2 Report
This paper reports on the chemistry of rectal gland secretions and headspace volatiles emitted by male and female Bactrocera kraussi (Hardy) (Tephritidae), and the detection of the identified compounds by gas chromatography-electroantennogram detection (GC-EAD) and -electropalpogram detection (GC-EPD). Various electrophysiologically active compounds were identified. Further behavioral assays are required to evaluate their potential applications in management of this important pest. The structure of this manuscript is well organized, and the manuscript is professionally prepared. However, instead of recommending direct acceptance of this manuscript in current form, I’d like to suggest some revisions to this high-quality work.
L28: Fig. 1 clearly show that compound 8 is also EAG responsive.
L92: Although presentation of their components in Table 1 is good, it would be better to provide a figure showing GC-MS chromatographs of headspace of volatiles of both sexes.
L128: it is critical to show time axis in this figure. Responsive indications in EAD traces of both sexes to both rectal gland extracts and EPD trace of females to female rectal gland extract are not so evident. Any possibility to increase scale of amplitude to show larger electrophysiological responses? The ratio of signal to noise is not bad in these recordings.
L148: delete “an”
L180: is it appropriate to say “interactions” of compounds with GC column?
